# Dopamine-Depleted Dopamine Transporter Knockout (DDD) Mice: Dyskinesia with L-DOPA and Dopamine D1 Agonists

**DOI:** 10.3390/biom13111658

**Published:** 2023-11-17

**Authors:** Vladimir M. Pogorelov, Michael L. Martini, Jian Jin, William C. Wetsel, Marc G. Caron

**Affiliations:** 1Department of Psychiatry and Behavioral Sciences, Duke University Medical Center, 354 Sands Building, 303 Research Drive, Durham, NC 27710, USA; 2Mount Sinai Center for Therapeutics Discovery, Departments of Pharmacological Sciences and Oncological Sciences, Icahn School of Medicine at Mount Sinai, New York, NY 10029, USA; michael.martini@mayo.edu (M.L.M.); jian.jin@mssm.edu (J.J.); 3Department of Cell Biology, Duke University Medical Center, Durham, NC 27710, USA; marc.caron@duke.edu; 4Department of Neurobiology, Duke University Medical Center, Durham, NC 27710, USA

**Keywords:** Parkinson’s disease, L-DOPA, dyskinesia, dopamine, dopamine transporter knockout mice, dopamine D1 agonists

## Abstract

L-DOPA is the mainstay of treatment for Parkinson’s disease (PD). However, over time this drug can produce dyskinesia. A useful acute PD model for screening novel compounds for anti-parkinsonian and L-DOPA-induced dyskinesia (LID) are dopamine-depleted dopamine-transporter KO (DDD) mice. Treatment with α-methyl-*para*-tyrosine rapidly depletes their brain stores of DA and renders them akinetic. During sensitization in the open field (OF), their locomotion declines as vertical activities increase and upon encountering a wall they stand on one leg or tail and engage in climbing behavior termed “three-paw dyskinesia”. We have hypothesized that L-DOPA induces a stereotypic activation of locomotion in DDD mice, where they are unable to alter the course of their locomotion, and upon encountering walls engage in “three-paw dyskinesia” as reflected in vertical counts or beam-breaks. The purpose of our studies was to identify a valid index of LID in DDD mice that met three criteria: (a) sensitization with repeated L-DOPA administration, (b) insensitivity to a change in the test context, and (c) stimulatory or inhibitory responses to dopamine D1 receptor agonists (5 mg/kg SKF81297; 5 and 10 mg/kg MLM55-38, a novel compound) and amantadine (45 mg/kg), respectively. Responses were compared between the OF and a circular maze (CM) that did not hinder locomotion. We found vertical counts and climbing were specific for testing in the OF, while oral stereotypies were sensitized to L-DOPA in both the OF and CM and responded to D1R agonists and amantadine. Hence, in DDD mice oral stereotypies should be used as an index of LID in screening compounds for PD.

## 1. Introduction

Parkinson’s disease (PD) is a heterogeneous neurodegenerative disorder with a varying age of onset, rate of progression, and symptoms. In the United States, PD affects approximately 1% of the population aged 55 years and older [1]. In 2020, it had a prevalence of 1 million people, and it has been projected that 1.2 million cases will be affected by 2030. Currently, the etiology of the disease is unknown; however, PD is associated with the loss of dopamine (DA) neurons, primarily in the substantia nigra pars compacta, and it is often accompanied with Lewy bodies and Lewy neurites containing α-synuclein [2,3]. These patients present with cardinal motor symptoms that include resting tremor, rigidity, bradykinesia, akinesia, facial masking, abnormal posture, gait anomalies, and difficulty initiating movement. Non-motor features may precede, occur with, or appear after the motor disturbances and these may involve anosmia, gastrointestinal difficulties, anxiety, depression, sleep disturbances, and cognitive decline [4].

While most PD cases are idiopathic, approximately 10% of cases are of monogenic origin with multiple contributing genetic and environmental risk factors [5]. A number of genetic mutations have been found. In familial PD, linkage analyses have been used to identify disease-causing mutations, whereas idiopathic PD risk factors have been reported in association analyses using PD patients and normal controls [6]. The genes most often associated with PD are those for α-synuclein (*SNCA*) and leucine-rich repeat kinase 2 (*LRRK2*), with less prevalent mutations in Parkin (*PRKN*), PTEN induced kinase 1 (*PINK1*), Parkinson disease gene 7 (*DJ-1*), ubiquitin carboxyl-terminal hydrolase isozyme L1 (*UCH-L1*), and ATPase cation transporting 13A2 gene (*ATP13A2*) [7,8].

Besides genetic factors, environmental factors are also implicated in the susceptibility to PD. These may involve exposure to the pesticide rotenone, the herbicide paraquat, the fungicide maneb, as well as chlorinated organic solvents and various metals [9,10,11,12,13,14,15]. However, the association between exposure to these agents and the risk for PD has been difficult to establish. Despite this fact, exposure to 1-methyl-4-phenyl-1,2,3,6-tetrahydropyridine (MPTP) is well known to produce PD symptoms [16]. To date, while many genetic and environmental risk factors have been identified for PD, the precise idiopathic basis of the disorder is unknown.

Due to the genetic associations found in PD patients, a number of animal genetic, viral-vector-mediated, and preformed viral models of the disease have been developed to recapitulate specific molecular pathologies linked to the candidate gene under study [17]. As an alternative approach, some investigators have developed models where the genes for transcription factors that regulate the birth and maintenance of DA neurons are disrupted [18,19,20,21,22]. These genetic models have been instrumental in discovering new molecular pathways for neurodegeneration or susceptibility to motor impairment. However, the models typically require the mice to age to develop the phenotype, where they may not be the most parsimonious models to use in rapidly screening novel compounds for treating PD.

To better mimic the progressive loss of DA neurons, the most prominent models are MPTP-treated monkeys [23,24] and mice [25,26], as well as 6-hydroxydopamine (6-OHDA) -lesioned rats [27] and mice [28]. MPTP-induced lesions in non-human primates are the highly preferred model that can show progressive motor deficits and, depending upon the frequency of administration, can produce α-synuclein inclusions [29,30]. Nevertheless, in both monkeys [31] and mice [32,33] the MPTP-induced motor deficits persist for variable intervals and in mice these effects are strain specific. By contrast, in the 6-OHDA model injections are administered unilaterally into the substantia nigra pars compacta, striatum, or medial forebrain bundle such that the non-lesioned ipsilateral limbs can serve as a control [34]. In this model, asymmetric motor abnormalities and motor rotations can be used to evaluate L-DOPA-induced sensitization and dyskinesia [34,35]. By comparison, bilateral 6-OHDA lesions typically result in severe aphagia and adipsia that may be accompanied by seizures and death [36,37]. Unfortunately, 6-OHDA-induced lesions fail to produce the gradual degeneration of DA neurons and the full neuropathology observed in PD patients.

A model that appears particularly amenable to screening novel compounds and studying signal transduction pathways for PD therapies is the DA transporter (DAT) knockout (KO) mouse [38,39,40,41,42,43]. In this model, when DAT mice are given α-methyl-*para*-tyrosine (AMPT) to block the conversion of L-tyrosine to L-DOPA, the levels of brain extracellular DA decline very rapidly. This result is due to the inability of DA neurons to retrieve DA from the synapse due to the loss of *Dat1* [44]. Hence, any DA used for neurotransmission has to rely entirely upon endogenous synthesis from L-tyrosine. Accordingly, these mice are termed DA-depleted DAT-KO (DDD) mice [38].

Within minutes after acute AMPT administration, DDD mice become akinetic; additional symptoms include catalepsy, tremor, and ptosis [38]. High doses of amphetamine derivatives alleviate the akinesia and catalepsy in these mice. Since the *Dat1* is deleted and DA synthesis is blocked with AMPT in the DDD mice, the amphetamine analogs appear to act through a non-DA mechanism. Importantly, acute administration of 3,4-dihydroxy-L-phenylalanine (L-DOPA) produces dramatic motor activation in DDD mice and chronic treatment leads to sensitization and behaviors likened to dyskinesia [42].

In PD patients, L-DOPA remains the first choice for therapy for treating their motor symptoms. However, long-term treatment with this DA precursor results in irregularities of motor symptom improvement (“on/off” phenomena), tolerance, and the emergence of abnormal involuntary movements termed “dyskinesia” [45,46]. A clinical feature of L-DOPA-induced dyskinesia (LID) is that it often becomes intractable and dissociates from the therapeutic effects of L-DOPA over time [47]. The hypothetical mechanism for these L-DOPA complications is postulated to involve unregulated DA release in PD brains [45,46]. When a significant number of DA neuron terminals are lost in PD, the remaining DA neurons can still secrete some DA [48,49,50]. This DA can be removed from the extracellular space by the norepinephrine and serotonin transporters [51]. L-DOPA can also be taken up by non-catecholaminergic neurons in the basal ganglia where it is converted to DA. This DA can be released in the absence of the DAT in the intact striatum. Alternatively, there is some evidence that L-DOPA can serve as a transmitter on its own at DA D1 receptors (D1Rs) [52].

When DDD mice are treated daily with L-DOPA and benserazide (Benz), locomotion in the open field (OF) gradually declines as vertical activities increase [42]. This latter activity is termed “three-paw dyskinesia” due to the characteristic pattern of standing on one leg while climbing the wall with three paws. These climbing movements against the wall have been classified as “abnormal involuntary movements” in other PD models [53,54]. Because this “dyskinesia” intensifies with chronic L-DOPA treatment and continues uninterrupted, it can be quantified as vertical beam-breaks in the OF [42]. Apart from the “three-paw dyskinesia”, the DDD mice also display tongue protrusions reminiscent of “orolingual abnormal movements” quantified as dyskinesia in the 6-OHDA mouse model [28].

We have hypothesized that in DDD mice L-DOPA induces a stereotypic activation of locomotion so intense the animals are unable to alter the course of their activities. Thus, upon encountering the walls of the apparatus, they will engage in “three-paw dyskinesia”. Consequently, if the vertical counts reflect this “dyskinesia”, then a drug or compound that decreases stereotypic locomotion will also reduce vertical counts, thereby resulting in a false positive for the anti-LID effect. To test this idea, we have compared vertical beam-breaks in the OF with that in a circular maze (CM) that does not obstruct horizontal movements. We also scored additional behaviors in these two test contexts (Table 1). Our objective was to establish a valid behavioral index of LID in the DDD mice based upon three criteria: sensitization to repeated L-DOPA administration, insensitivity to the test context, and response to direct DA receptor stimulation (increase) or to amantadine (decrease), the latter being clinically effective in reducing symptoms of dyskinesia in PD patients [55].

For direct DA receptor stimulation, D1R agonists were used. Notably, D1Rs are targets for anti-parkinsonian L-DOPA treatment [56] as activation of this receptor can alleviate parkinsonism in MPTP-treated monkeys [23]. Similarly, a study with PD patients has reported beneficial effects of a D1R agonist on motor function and alertness [57]. However, D1R agonists contribute to LID since deletion of D1Rs, but not D2Rs, blocks the presentation of LID in 6-OHDA-lesioned mice [58]. In the present study, the full D1R agonists SKF81297 and MLM55-38 (i.e., compound 10 in [59]) were used. MLM55-38 is a novel non-catechol ligand with potent D1R agonist activity and balanced G-protein and β-arrestin signaling similar to DA. Both compounds were tested for LID effects (i.e., with L-DOPA) and for anti-parkinsonian/pro-dyskinetic effects after L-DOPA was withdrawn.

## 2. Materials and Methods

### 2.1. Animals

Adult (5–9 months old) male DAT-KO mice were acquired from Dr. Marc Caron’s lab. Animals were housed 3–5 per cage on a 14:10 h light:dark cycle (lights on at 0700 h) in a humidity- and temperature-controlled room, with food and water provided *ad libitum*. The behavioral experiments were conducted from 0900–1400 h. All experiments were conducted with protocols (A071-17-03 and A077-20-03) approved by the Institutional Animal Care and Use Committee at Duke University and according to ARRIVE guidelines.

### 2.2. Drugs

Amantadine, AMPT, L-3,4-dihydroxyphenylalanine-HBr (L-DOPA), Benz (Sigma-Aldrich, St. Louis, MO, USA), and SKF81297 (Bio-Techne Corporation, Minneapolis, MN, USA) were dissolved in sterile saline (Hospira Inc, Lake Forest, IL, USA). Compound MLM55-38 [59] was dissolved in 5% N-methyl-2-pyrrolidone (Sigma-Aldrich), 5% Kolliphor (Sigma-Aldrich), and 90% sterile saline (Hospira). All compounds were injected i.p. except L-DOPA/Benz which were given s.c. All injection volumes were 5 mL/kg.

### 2.3. Apparati

Locomotion and dyskinesia-like behaviors were evaluated in a 42 × 42 × 30 cm OF (Omnitech Inc., Columbus, OH, USA) and a circular maze (CM). The CM was composed of two circular Plexiglas walls affixed to a flat Plexiglas base: the inner and outer walls had diameters of 20 and 40 cm, respectively, and were 20 cm high. These walls formed an inner circular arena and an outer circular runway (bounded by the inner and outer rings); the latter comprised the CM. The apparatus was placed into the OF such that horizontal and vertical motor activities within the circular areas could be monitored by infrared diodes interfaced to a computer with Fusion 5.3 software (Omnitech Inc.). The lighting level in arenas was 180 Lux. No aversive stimuli were used during the tests. The activities were converted to distance traveled and vertical activity counts, respectively. Stereotyped rearing, climbing, and oral stereotypies in DDD mice (Table 1) were scored “live”.

### 2.4. Procedures

The experimental design is presented in Figure 1. Mice were always placed into the test room at least 20 min before treatment. As in Urs and colleagues [42], mice received 125 mg/kg AMPT in their home cages, followed 60 min later with vehicle or 25 mg/kg L-DOPA plus 12.5 mg/kg Benz in the same injection; the mice were tested immediately. Motor activities were recorded automatically and other behaviors were filmed over 60 min or scored “live”. Note, AMPT blocks the synthesis of DA and renders DAT-KO mice cataleptic [38].

To induce dyskinesia in the OF or CM, DDD mice were administered 125 mg/kg AMPT plus 25/12.5 mg/kg L-DOPA/Benz and tested in the OF or CM for 13 days (Figure 1). To determine whether the test context was important for the expression of behaviors, a subset of randomly selected AMPT plus L-DOPA/Benz mice had their OF and CM contexts reversed on days 15 and 17. All mice continued to receive AMPT plus L-DOPA/Benz treatments during this time.

As 25 mg/kg L-DOPA induced intense stereotypies in DDD mice, the effects of D1R agonists were tested with 125 mg/kg AMPT plus 6/12.5 mg/kg L-DOPA/Benz in the OF and CM, such that the agonist effects could be more clearly observed (Figure 1). Following AMPT administration, the vehicle, 5 mg/kg SKF81297, or 5 or 10 mg/kg MLM55-38 was given immediately prior to injecting L-DOPA/Benz and the mice were tested in the OF or CM. After a 7-day washout period of agonists and L-DOPA/Benz, AMPT-treated animals were tested in the OF with the vehicle, SKF81297, or both doses of MLM55-38 (Figure 1) to assess the anti-parkinsonian or pro-dyskinetic potential of the compounds.

Subsequently, the mice were re-sensitized with 125 mg/kg AMPT plus 25/12.5 mg/kg L-DOPA/Benz for 4 days and the effects of amantadine in reducing LID-like behaviors were examined (Figure 1). Here, the AMPT was given and this was followed 10 min later with vehicle or 45 mg/kg amantadine that was administered 50 min before receipt of L-DOPA/Benz.

### 2.5. Statistics

The results are presented as means ± SEMs. All statistics were performed with SPSS 25 (IBM-SPSS, Chicago, IL, USA). Univariate ANOVAs evaluated treatment effects. Two-way ANOVA examined test context during sensitization and context reversal, test context and D1R or amantadine treatments. Repeated measures ANOVA evaluated changes over time as the “within subjects” factor and treatment as the “between subjects” factor. *Post hoc* analyses were Bonferroni-corrected pair-wise comparisons, except when Dunnett tests compared drug responses to vehicle. Violations of homogeneity of variance were assessed with Levene’s test and were followed by Welch’s one-way ANOVA with Games–Howell *post hoc* tests or with Greenhouse–Geisser corrections for RMANOVA. A *p* < 0.05 was considered statistically significant.

## 3. Results

### 3.1. L-DOPA Sensitization in DDD Mice

DDD mice were treated with 125 mg/kg AMPT and 25/12.5 mg/kg L-DOPA/Benz (see [41]) over 17 consecutive days (Figure 1). Locomotion declined in mice in both the OF and CM over the first 13 days (*p*-values ≤ 0.001) (Figure 2A). Vertical activity counts were higher in the OF than CM on days 9–13 (*p*-values ≤ 0.016) (Figure 2B). Within the OF, vertical counts increased from days 1 to 3 (*p* = 0.038) and remained high to day 13, whereas in the CM they increased initially then declined. An examination of supported rearing revealed this response was higher across time in the CM than OF beginning on day 3 to day 13 (*p*-values ≤ 0.045) (Figure 2C). No changes in supported rearing were observed in the CM after day 3, while in the OF this response declined from a high on day 1 to a low on days 7 and 13 (*p*-values ≤ 0.050). By contrast, relative to the CM climbing increased in the OF from day 3 to day 13 (*p*-values ≤ 0.006) (Figure 2D). This augmentation in climbing within the OF was evident from day 3 (*p* = 0.005) and it continued to increase through day 13 (*p*-values < 0.001), whereas no changes occurred in the CM over the same period. Thus, climbing reflects L-DOPA-induced sensitization of vertical activity in the OF, whereas this response is largely absent in the CM.

Besides changes in vertical stereotypies, oral stereotypies showed similar amounts of sensitization in both test contexts across days (*p* < 0.001) (Figure 2E). Collectively, climbing and oral stereotypies show L-DOPA-induced sensitization in the OF. However, climbing is largely absent in the CM, while oral stereotypies are undifferentiated between the two test contexts. Hence, climbing appears context dependent, whereas oral stereotypies do not show this dependency.

### 3.2. Changes in Test Context during L-DOPA Sensitization in DDD Mice

To determine whether the test context was important for the expression of behaviors, a random subset of DDD mice had their OF and CM contexts reversed on days 15 and 17 (Figure 1). Here, responses on the last day of sensitization (day 13) were compared to those during reversal days. Animals originally in the OF on day 13 increased their locomotion when placed into the CM on days 15 and 17 (*p*-values < 0.001) (Figure 2A, *arrow*). By contrast, CM mice failed to increase their activities when changed to the OF. For vertical activity counts, this response was reduced in the interchange from the OF to CM on both days (*p*-values ≤ 0.006) (Figure 2B, *arrow*); the converse was seen in the CM to OF transition (*p*-values ≤ 0.036). Supported rearing was enhanced in changes both from the OF to CM and *vice versa* from day 13 to days 15–17 (*p* = 0.049) (Figure 2C, *arrow*). Climbing decreased from the OF to the CM (*p* < 0.001) and it increased in the converse transition (*p* = 0.001) (Figure 2D, *arrow*). Incidences of oral stereotypies were similar in both test contexts on day 13 and reversal (Figure 2E, *arrow*). Collectively, these results provide further support that the test context influences the expression of climbing in the L-DOPA-sensitized DDD mice, whereas it exerts no effects on oral stereotypies.

### 3.3. Effects of D1R Agonists on Stereotypic Behaviors in DDD Mice in Two Test Contexts

Because 25 mg/kg L-DOPA induced intense stereotypies in DDD mice, a lower 6 mg/kg dose of L-DOPA was used, so that possible agonist responses would not be obscured. Here, effects of the vehicle and D1R agonists (SKF81297 and MLM55-38) were compared in the OF and CM with DDD mice (Figure 1). Locomotion was increased with both D1R agonists compared to L-DOPA/Benz with vehicle in the OF, but this enhancement was only significant with 5 mg/kg MLM55-38 in the CM (*p* = 0.041) (Figure 3A,B). Vertical activity counts were increased only in the OF with SKF81297 (*p* = 0.005) and both doses of MLM55-38 (*p*-values ≤ 0.023) (Figure 3C); they were unchanged in the CM (Figure 3D). Supported rearing was increased in the OF with SKF81297 (*p* = 0.022) and both doses of MLM55-38 (*p*-values ≤ 0.031) (Figure 3E), while only the two doses of MLM55-38 stimulated this response in the CM (*p*-values ≤ 0.006) (Figure 3F). Incidences of climbing were low in both settings (Figure 3G,H) but were increased in the CM by 10 mg/kg of MLM55-38 (*p* = 0.019) (Figure 3H). Further stereotypy studies revealed that oral stereotypies in the OF were augmented by SKF81297 and 10 mg/kg MLM55-38 *versus* the vehicle (*p*-values ≤ 0.036), while in the CM all drug treatments stimulated these responses (*p*-values ≤ 0.004) (Appendix A). Collectively, these findings show that with this low 6 mg/kg L-DOPA dose, test context can influence responses to D1R agonists. Moreover, it appears that climbing is low and variable across treatments and is not a major contributor to vertical activity in this experiment in either context. By comparison, oral stereotypies are stimulated by the D1R agonists in both test contexts.

### 3.4. Effects of D1R Compounds in the DDD Model without L-DOPA

To assess further the anti-parkinsonian/pro-dyskinetic potential of the compounds, sensitized DDD mice were given a 7-day washout period from the compounds and L-DOPA/Benz and responses to the vehicle, SKF81297, or MLM55-38 were tested in the OF with AMPT (Figure 1). Both doses of MLM55-38 stimulated locomotion relative to vehicle (*p*-values ≤ 0.003) and SKF was more efficacious than 5 mg/kg MLM55-38 (*p* = 0.003) with a strong trend relative to the 10 mg/kg dose (*p* = 0.051) (Appendix A). None of the D1R agonists influenced vertical activity counts, supported rearing, climbing, or oral stereotypies. Thus, pro-dyskinetic responses to D1R agonists were not detected in DDD mice when L-DOPA/Benz treatments were absent.

### 3.5. Effects of Amantadine in the DDD Mice Treated with L-DOPA

Amantadine is clinically effective in reducing symptoms of dyskinesia in PD patients [55]. DDD mice were re-sensitized for 4 days with 125 mg/kg AMPT and 25/12.5 mg/kg L-DOPA/Benz DDD mice (Figure 1). This L-DOPA dose was selected so that any reductions in behavior with amantadine could be detected. Following sensitization, the mice received AMPT 10 min prior to administering the vehicle or 45 mg/kg amantadine followed 50 min later by L-DOPA/Benz. DDD mice were placed immediately into the OF or CM and tested. Compared to vehicle, amantadine reduced overall locomotor activity (*p* < 0.001) (Figure 4A). Vertical activity counts in the vehicle-treated group were higher in the OF than CM (*p* = 0.002) and amantadine non-significantly reduced this activity in the OF and increased it in the CM (Figure 4B). Supported rearing was significantly higher overall in the CM than OF (*p* = 0.031) (Figure 4C). Climbing was higher overall in the OF than CM (*p* < 0.001), and amantadine decreased this behavior in both test contexts (*p* = 0.006) (Figure 4D). Oral stereotypies were also reduced with amantadine in both test contexts (*p* = 0.010) (Figure 4E). Together, these findings show that amantadine decreases locomotor activities, while vertical counts reflect different behaviors in OF and CM that are responsive to amantadine. Climbing is reduced, while rearing is enhanced. Since oral stereotypies are decreased irrespective of the test context, they appear to be a valid index of LID in DDD mice.

## 4. Discussion

No single animal model can fully reproduce the PD pathophysiology and symptoms. For instance, genetic models of PD are useful in mimicking some of the pathophysiological mechanisms of the disorder, while loss of DA neurons may not be as pronounced as in PD patients [60]. By comparison, toxin-based models have face validity for the primary deficit in DA neuronal loss, but often lack the pathological features of the disease [26,31]. By contrast, DDD mice are, at best, an acute model of PD. Here, DDD mice represent a model of dramatic brain-wide DA deficiency that can be achieved rapidly, consistently, and with little variability among mice [38,39]. In this way, DA levels can be depleted precisely such that stereotypic behavior—likened to LID—can emerge.

### 4.1. Sensitization in DDD Mice

Dyskinesia in PD arises as a side-effect of long-term exposure to L-DOPA or DA agonists [46,47,55]. Its expression requires two conditions: loss of DA neurons in the nigro-striatal pathway and prolonged treatment with L-DOPA [61]. In patients, it can take years for LID to develop [46] and it often presents as akhatisia, ballism, and chorea [47,62]. By comparison, LID takes days to weeks to develop in MPTP-treated non-human primates and in 6-OHDA-lesioned rodents with L-DOPA administration [30,31,32,61,63]. While the presentation of LID is difficult to model in animals, the MPTP model in non-human primates comes the closest in terms of face validity to the LID ratings used for PD patients [23,29]. By contrast, 6-OHDA-lesioned rodents develop sensitization to L-DOPA over a short period such that “abnormal involuntary movements” emerge that include “repeated jerking”, “circular forelimb movements”, or “forepaw moving in a lateral direction” [63,64,65].

With respect to DDD mice, L-DOPA-induced vertical activity as “three-paw dyskinesia” gradually increases over days before plateauing [42]. In the present study, vertical counts, an index of LID, reached a plateau by day 3 of L-DOPA administration in the OF, while climbing stereotypy showed a more gradual increase over 9 days. Thus, relative to vertical counts, climbing stereotypy appears to be a more precise index of LID sensitization in this test context. Another form of stereotypy in DDD mice showing sensitization was oral stereotypy. This behavior also reached a maximum in 9 days with L-DOPA treatment in both the OF and CM. Thus, sensitization of stereotypies in DDD mice has certain parallels with LID indices found in other classic animal models of PD [23,28,29,35,36]. Besides sensitization over time, the identification of LID in the DDD mice should meet additional criteria to be considered valid.

### 4.2. Stereotypy and LID in the DDD Mice during Changes in the Test Context

An issue in DDD mice is the relationship between horizontal and vertical activities and its modulation by the test context. Non-treated DAT-KO mice are known to display intense horizontal activities and thigmotaxis in the OF [66]. However, reducing the size of the OF augments their vertical activity, in the form of stereotypical rearing, due to constraints on locomotion [66]. To examine whether forward locomotion in the OF with DDD mice treated with L-DOPA contributes to climbing (i.e., “three-paw dyskinesia”), we sensitized the DDD mice in a different test context—the CM. Since this apparatus does not have corners, it permits unobstructed locomotion. When DDD responses were examined in the CM, vertical counts and climbing stereotypy were low compared to the OF. Importantly, the interchange from the OF to CM and *vice versa* revealed the increased vertical activity and climbing stereotypy depended upon the test environment. A similar relationship between locomotor and vertical activities has been reported in reserpine-treated rats [67]. By comparison, the oral stereotypies of DDD mice in the OF and CM did not conform to this test context dependence.

### 4.3. Responses to D1R Agonists and Amantadine by DDD Mice

If an index of LID depends upon the test context, then the change in the response to drugs may reflect the interaction between motor activity and the test apparatus. In the case of the DDD model, a motor-impairing effect of a drug or compound may be manifested as a reduction in vertical counts. To further investigate this potential confound, we tested compounds known to induce dyskinesia or to oppose it in the clinic [68,69,70].

D1R agonists produced anti-PD effects and dyskinesia similar to L-DOPA in advanced and early-stage PD [68,69] and in parkinsonian non-human primates [71]. To test the effects of D1R agonists on behaviors in DDD mice, a lower 6 mg/kg L-DOPA dose was provided so the potentiation of LID could be seen more readily. SKF81297 and MLM38-55 increased locomotion in CM and vertical activity in OF, in accordance with our hypothesis. Notably, climbing was replaced with rearing in these conditions. By contrast, when L-DOPA was omitted, the D1R agonists failed to induce vertical activity or climbing. This result confirmed earlier reports on insensitivity of the model to DA agonists [38]. With regard to oral stereotypy, SKF and MLM38-55 stimulated this response to similar extents in both test contexts in the presence of L-DOPA, further supporting it as an index of LID in these mice.

Besides the D1R agonists, DDD mice were tested with amantadine, a weak non-competitive NMDA receptor antagonist and DAT inhibitor [70]. Amantadine is clinically effective in reducing symptoms of dyskinesia in PD patients [55]. In our experiment, DDD mice were returned to the high 25 mg/kg dose of L-DOPA so that any behavior-reducing effects could be observed. When amantadine was administered in the OF, locomotion, vertical counts, and climbing declined. This finding indicates an effect on locomotion rather than on dyskinesia since horizontal and vertical activities in DDD mice should dissociate instead of representing a change in overall motor activities. By contrast, in the CM locomotion declined while vertical activity did not. This latter finding confirms our hypothesis that vertical counts as an index of LID are confounded with locomotion. With respect to oral stereotypies, amantadine decreased this behavior in both the OF and CM. Thus, oral stereotypy is sensitized to L-DOPA, responds to D1R agonists and amantadine, and shows insensitivity to the test context. For this reason, oral stereotypies appear to be the most valid index of LID in the DDD model.

## 5. Conclusions

Similarities among behaviors ascribed to PD and those responsive to pharmacological treatments are essential for demonstrating the efficacy of experimental compounds in preclinical models [72]. Our studies emphasize the importance of direct observation of mouse behaviors and suggest three criteria for identifying “dyskinetic-like” behavior: sensitization to repeated L-DOPA administration, sensitivity to D1R agonists and amantadine, and independence of the test context. Oral stereotypies conform to these criteria and should be used as a valid measure of dyskinesia when using DDD mice to test novel compounds that may possess anti-parkinsonian effects and/or serve to reduce LID.

## Figures and Tables

**Figure 1 biomolecules-13-01658-f001:**
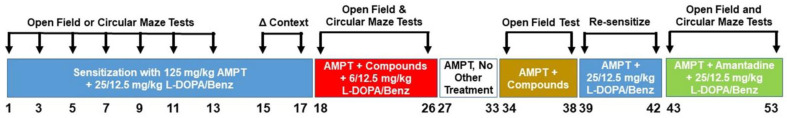
Experimental design for studies with DDD mice. The arrows denote the days or sessions when mice were tested. The information within the boxes describes the treatments. The numbers below the boxes indicate the days on which the mice received a certain treatment and test.

**Figure 2 biomolecules-13-01658-f002:**
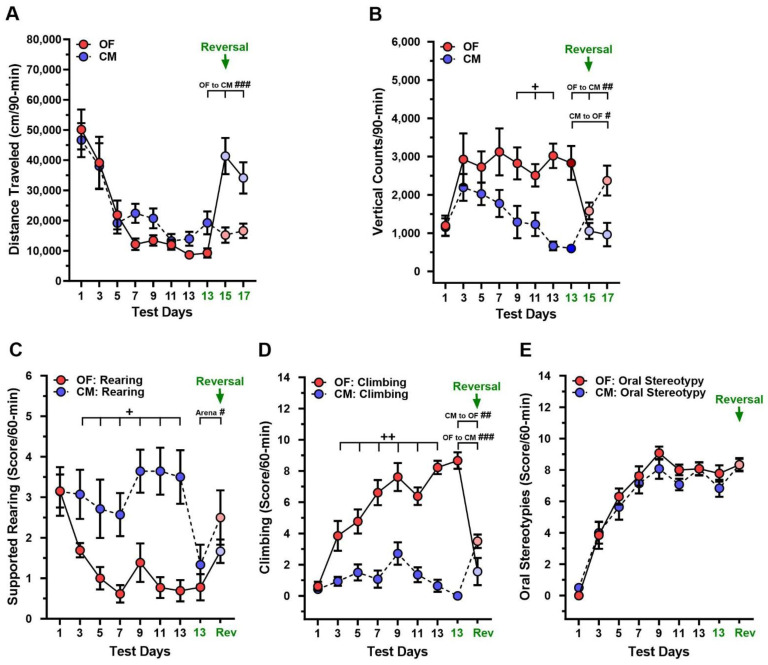
Responses to L-DOPA-induced sensitization in the open field and circular maze with dopamine-depleted dopamine transporter knockout (DDD) mice. Sensitization over days and reversal of test contexts with DDD mice treated daily with 125 mg/kg AMPT and 25/12.5 mg/kg L-DOPA/Benz. DDD mice were sensitized to L-DOPA/Benz and tested in the open field (OF) or circular maze (CM) over 13 days (see Figure 1). The test context was reversed (*arrow*) from day 13 to days 15 and 17. Note, since context reversal was examined in a subset of mice, day 13 results are shown both with all mice at sensitization and in the subset of mice used in reversal (days 13–17). (**A**) Distance traveled RMANOVA (sensitization, days 1–13): time (F(6,150) = 23.807, *p* < 0.001); RMANOVA (reversal, days 13 vs. 15–17): time (F(2,26) = 8.412, *p* = 0.002), time by apparatus (F(2,26) = 13.696, *p* < 0.001), and apparatus (F(1,13) = 5.860, *p* < 0.001). (**B**) Vertical activity counts RMANOVA (sensitization, days 1–13): time (F(6,150) = 4.700, *p* < 0.001), time by apparatus (F(6,150) = 2.798, *p* = 0.013), and apparatus (F(1,25) = 10.265, *p* = 0.004); RMANOVA (reversal, days 13 vs. 15–17): time by apparatus (F(2,26) = 19.450, *p* < 0.001). (**C**) Supported rearing RMANOVA (sensitization, days 1–13): time (F(6,150) = 2.379, *p* = 0.032), time by apparatus (F(6,150) = 2.311, *p* = 0.037), and apparatus (F(1,25) = 26.990, *p* < 0.001). RMANOVA (reversal, days 13 vs. 15–17): apparatus (F(1,13) = 4.738, *p* = 0.049). (**D**) Climbing RMANOVA (sensitization, days 1–13): time (F(6,150) = 19.611, *p* < 0.001), time by apparatus (F(6,150) = 11.617, *p* < 0.001), and apparatus (F(1,25) = 58.443, *p* < 0.001); RMANOVA (reversal, days 13 vs. 15–17): time (F(1,13) = 10.452, *p* = 0.007), time by apparatus interaction (F(1,13) = 90.248, *p* < 0.001), and apparatus (F(1,13) = 20.927, *p* < 0.001). (**E**) Oral stereotypies RMANOVA (sensitization, days 1–13): time (F(6,150) = 72.343, *p* < 0.001); RMANOVA (reversal, days 13 vs. 15–17): time (F(1,13) = 5.864, *p* = 0.031). For sensitization N = 13 (OF) and N = 14 (CM) for panels A–E; for reversal N = 9 (OF) and N = 6 (CM) for panels A–E. The data are presented as means ± SEMs; *post hoc* tests were Bonferroni comparisons. ^+^
*p* < 0.05, ^++^
*p* < 0.01, OF vs. CM; ^#^
*p* < 0.05, ^##^
*p* < 0.01, ^###^
*p* < 0.001, day 13 vs. days 15 and 17 (reversal) from OF or CM to *vice versa*.

**Figure 3 biomolecules-13-01658-f003:**
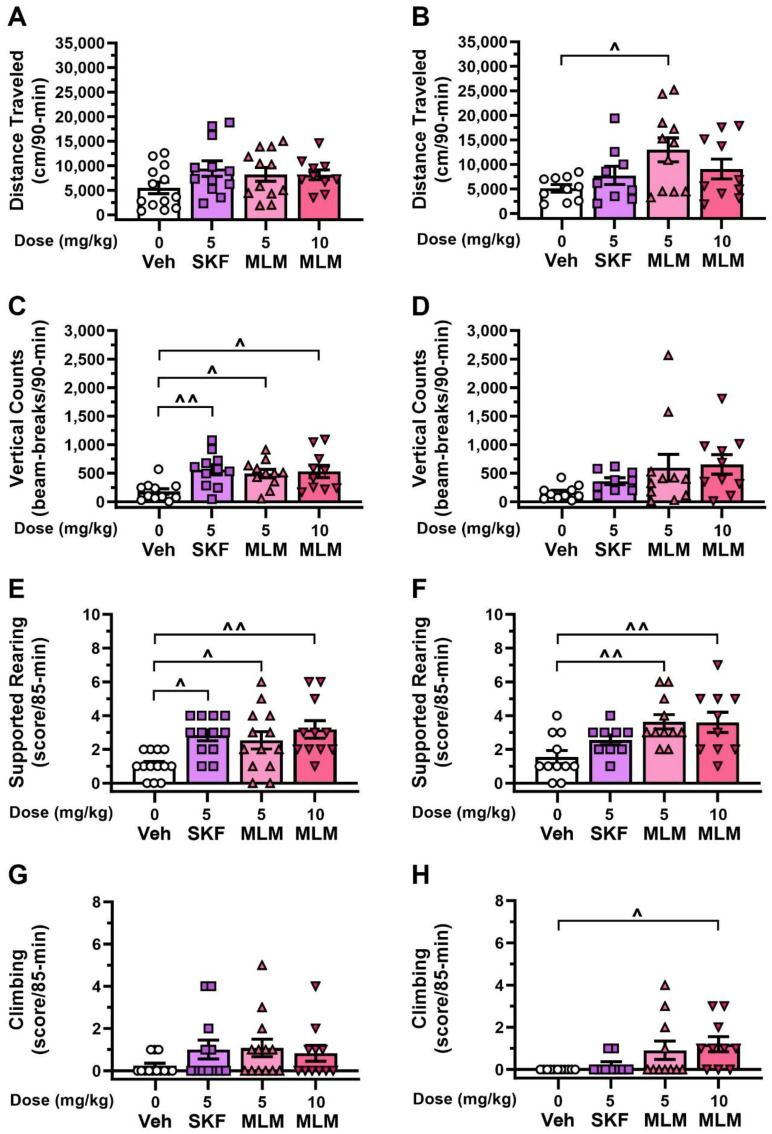
Behavioral responses to dopamine D1 receptor agonists in L-DOPA-sensitized dopamine-depleted dopamine transporter knockout (DDD) mice. DDD mice were given 125 mg/kg AMPT treated with compounds and 6/12.5 mg/kg L-DOPA/Benz and tested in the open field (OF) (*left*) or circular maze (CM) (*right*) (see Figure 1). Note, in this experiment (panels (**A**–**D**), data were lost for 1 mouse treated with 10 mg/kg MLM in the OF and 1 vehicle-treated mouse in the CM for distance traveled and vertical activity counts. (**A**,**B**) Distance traveled ANOVA for OF: not significant; Welch ANOVA for CM: treatment (F(3,17.703) = 3.865, *p* = 0.027). (**C**,**D**) Vertical activity counts ANOVA for OF: treatment (F(3,39) = 4.798, *p* = 0.006); Welch ANOVA for CM: treatment (F(3,18.037) = 5.068, *p* = 0.010), Games–Howell *post hoc* tests were not significant. (**E**,**F**) Supported rearing ANOVA for OF: treatment (F(3,40) = 4.511, *p* = 0.008); ANOVA for CM: treatment (F(3,37) = 5.186, *p* = 0.004). (**G**,**H**) Climbing stereotypy ANOVA for OF: not significant; ANOVA for CM: treatment (F(3,37) = 3.579, *p* = 0.023). Abbreviations: Veh, vehicle; SKF, SKF81297; MLM, MLM55-38. N = 10–11 mice/group (OF); N = 9–11 mice/group (CM). The data are presented as means ± SEMs; *post hoc* tests were Dunnett comparisons to vehicle. ^ *p* < 0.05, ^^ *p* < 0.01, Veh vs. other treatments.

**Figure 4 biomolecules-13-01658-f004:**
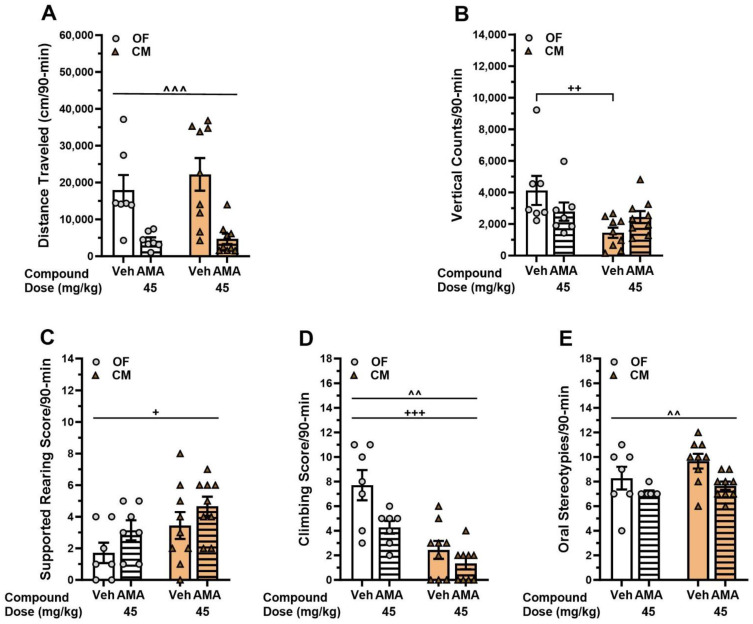
Effects of amantadine in DDD mice. DDD mice were re-sensitized over 4 days with 125 mg/kg AMPT and 25/12.5 mg/kg L-DOPA/Benz (Figure 1). Following sensitization, the mice received AMPT 10 min prior to administering the vehicle or 45 mg/kg amantadine (AMA) followed 50 min later by L-DOPA/Benz and were tested immediately in the open field (OF) or circular maze (CM). (**A**) Distance traveled two-way ANOVA: treatment (F(1,28) = 23.520, *p* < 0.001). (**B**) Vertical activity counts two-way ANOVA: apparatus (F(1,28) = 7.520, *p* = 0.011) and treatment by apparatus (F(1,28) = 4.469, *p* = 0.044). (**C**) Supported rearing two-way ANOVA: apparatus (F(1,28) = 5.188, *p* = 0.031). (**D**) Climbing two-way ANOVA: treatment (F(1,28) = 8.801, *p* = 0.006) and apparatus (F(1,28) = 28.872, *p* < 0.001). (**E**) Oral stereotypy two-way ANOVA: treatment (F(1,28) = 7.747, *p* = 0.010); N = 7–9 mice/treatment/test context. The data are presented as means ± SEMs; *post hoc* tests were Bonferroni comparisons. ^+^
*p* < 0.05, ^++^
*p* < 0.01, ^+++^
*p* < 0.001, OF vs. CM; ^^ *p* < 0.01, ^^^ *p* < 0.001, Veh vs. AMA.

**Table 1 biomolecules-13-01658-t001:** Behaviors scored in Dopamine-Depleted Dopamine transporter knockout mice.

Behavior ^a,b,c^	Description
Supported rearing	Standing on one or two hind legs with the forepaws on the wall
Climbing	Standing on the tail and/or one or two hind legs with the forepaws on the wall as if attempting to climb
Oral stereotypies	Licking and/or gnawing behaviors (i.e., opening and closing the lower jaw)

^a^ Each behavior was given a score of 1 if present or 0 if absent. ^b^ Most behavioral scores were summed over 12 observation periods for each animal over the 60 min test. ^c^ For compound testing, mice were subjected to 90 min sessions but were observed for 1 min beginning at the first 5 min and continuing at 10 min intervals so the maximum score was 9 over 85 min.

## Data Availability

The data are available upon reasonable request.

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
