# Peer review of "Dopamine-Depleted Dopamine Transporter Knockout (DDD) Mice: Dyskinesia with L-DOPA and Dopamine D1 Agonists"

_biomolecules, 2023, doi:10.3390/biom13111658_

Round 1

Reviewer 1 Report

Comments and Suggestions for Authors

This is an intriguing study in which the authors conducted a series of behavioural tests on mouse models to investigate L-Dopa-induced dyskinesia. The selected mouse models are well-suited for this study, and demonstrate significant differences in various tested motor functions and behavioural patterns. The figures are consistently well-presented and clear.

However, I found this manuscript difficult to read and follow. The purpose of the study or the rationale behind (e.g. using different mouse models), as well as the novelty and significance, is mentioned somehow but not adequately emphasised. This issue is particularly noticeable in the structure of the introduction and abstract. A more accurate and catchy title is suggested.

Given that this is primarily a behavioural study, it would be beneficial if the authors could establish clearer connections between their findings and clinical phenotypes in the discussion section, or mention future investigations into the underlying mechanisms.

Comments on the Quality of English Language

The scientific writing style could benefit from some improvements. For example, the author mentioned the "bArr model" and the hypothesis before explaining its function or its relevance to antiPD/LID. Additionally, the text contains repeated words (e.g., "parenthetically"). In the methods section, more precise terminology could be use, such as using "6-OHDA mice" instead of "6-OHDA animals", and specifying "intervention/procedure/drug delivery" instead of using a broad term like "surgery." 

Reviewer 2 Report

Comments and Suggestions for Authors

As stated in the title, authors aimed to provide a comparison of L-Dopa induced dyskinesia in DAT-KO mice that were depleted in dopamine and in bArr2-KO mice. As written, the main aim of this study is unclear to me. There are too many variables: 2 different Ko mice with no apparent relation between them, different pharmacological treatments in addition to 6-OHA injections, Dopamine depletion and L-Dopa treatment. As such, it is very difficult to follow this study and understand where it is going. What is the focus, what is the take home message, how the findings are relevant to Parkinson’s disease and of interest to the scientific community? The whole manuscript needs to be carefully edited, and a much better focus on the main aims and the corresponding results (and conclusions) need to be made before this is communicated to the scientific community.

Many sentences are too obscure, such as  “D1Rs are targets for Anti-Parkinson’s L-dopa treatment”. What does that even mean? All Dopamine receptors are target to L-dopa treatment, although not sure if this is obtained directly or not (i.e., whether L-dopa directly acts on DR or indirectly after their transformation to dopamine is to be established).

Many paragraphs in the result section end with a statement that seems too vague and even irrelevant. For instance:  

-       “The test context influences the expression of climbing in the L_dopa-sensitized DDD mice, whereas it exerts no effects on oral stereotypies “. I am not sure what to make of this, why is it significant to this study? 

-       Another example: “it appears that climbing is not a major contributor to vertical activity in this experiment in either context “. Again: why do I need to know this and how this relates to Parkinson’s disease and its management? 

-       And another one: The fact that vertical activates remained unchanged across all days indicates that beam-breaks do not differentiate between climbing and supported rearing”. What is this? Why is it relevant to anything?

-        And yet another one: “While some sex effects emerged in this experiment, the incidences of stereotypy and postural dyskinesia were more pronounced in barr2 KO than WTs over the first few days of treatment and then were lost”. And ? Why is this here and how does this help answer the initial question ?

Other comments:

-       Authors need to provide information regarding the means by which 6-OHA and L-Dopa enter dopamine neurons in the absence of the DAT. After all it was shown repeatedly that DAT Ko animals are totally immune to the neurotoxic effects of DA depleting compounds such as MPTP.

-       Authors need to provide more detailed info in the abstract section: name and dose used for D1R agonists and other pharmacological agents. They also need to provide in the abstract a clearer description of the rational of the study and its main conclusions with relevance to Parkinson’s disease.

-       The end sentence of the introduction is weird: there no strong statement as to the main aim of the study nor to the procedures used.

-  Many paragraphs of the introduction are actually summaries of the results.

-       Why studies were performed on male DAT-Ko mice and on male and female Barr2-ko ? 

-       Is there a specific number of approval for animal care and study ?

-       The procedures paragraph in the materials and methods section is 2 pages long ! This exemplifies the complexity of the study and clearly indicates that there is a dire need for re-writing and careful eidting.

-       There is an excessive use of the word “Parenthetically”…

Reviewer 3 Report

Comments and Suggestions for Authors

The study was conducted well, however, I may have a few queries/suggestions

1.       Authors may modify the title of the manuscript based on the hypothesis or findings. Looks generic. Objectives or findings seem to be missing

2.       The study approval number is missing from the manuscript.

3.       Authors may cite a reference for the surgery procedure with a suitable original article or protocol.

4.       For OF: The acclimatization condition of the animals before the behavioural assessment and lighting condition (Lux used), whether authors used any aversive stimuli are missing. Authors may explain these conditions. The author may cite a suitable ref. for the methodology.

5.       Line 145, routes of administration are missing for AMPT.

6.       Citations are very old, authors may try to replace them with appropriate new original articles.

Comments on the Quality of English Language

Needs editing and proofreading

Round 2

Reviewer 1 Report

Comments and Suggestions for Authors

The revised manuscript is clearer and easier to read.

Minor points:  

In the Intro and Abstract, there are long sentences (Line 20-23, 37-39, 72-75,128-130) that can be improved by adding commas.

Intro: Font size is not consistent.

Line287-288 need rephrase.

Comments on the Quality of English Language

/

Author Response

I have re-written the long sentences in the Abstract (lines 20-23) and Introduction (lines 37-39, 72-75, and 128-130). The information originally in lines 128-130 has been re-written and can be found in lines 130-132. I have also checked the font size in the Introduction and made it consistent throughout. I have re-phrased lines 287-288 (now lines 290-292 in the revised "galleys"). The title in the "galleys" was highlighted because it was changed.

Reviewer 2 Report

Comments and Suggestions for Authors

The authors have addressed my major concerns and the paper now is better focused and written. However, the take home message as summarized in the last sentence of the abstract is still weak : "In DDD mice, oral stereotypies should be used as an index of L-dopa Induced Dyskinesia in screening compounds for PD". As such the relevance of this study to the scientific community is still very marginal.

Author Response

Thanks